# The Physical Demands of NCAA Division I Women’s College Soccer

**DOI:** 10.3390/jfmk4040073

**Published:** 2019-12-12

**Authors:** Robert W. Sausaman, Matt L. Sams, Satoshi Mizuguchi, Brad H. DeWeese, Michael H. Stone

**Affiliations:** 1Athletic Performance Department, University of Missouri, Columbia, MO 65211, USA; 2Performance & Nutrition Department, Kansas City Royals, Surprise, AZ 85374, USA; matt.l.sams@gmail.com; 3Center of Excellence for Sport Science and Coaches Education, Department of Sport, Exercise, Recreation, and Kinesiology, East Tennessee State University, Johnson City, TN 37614, USA; mizuguchi@etsu.edu (S.M.); deweese@etsu.edu (B.H.D.); stonem@etsu.edu (M.H.S.)

**Keywords:** college soccer, GPS, physical demands, women’s soccer, physical match performance

## Abstract

Extensive research into women’s collegiate soccer is scarce, leaving gaps in the literature with little information available detailing the physical demands at different standards of play. Our purpose was to elucidate the physical demands of the Division I collegiate level and identify differences between playing positions. Twenty-three field players were observed during four competitive seasons using 10-Hz GPS units (Catapult Sports, Melbourne, Australia). Descriptive statistics and 95% confidence intervals were used to determine group and position-specific physical demands. Linear mixed modelling (LMM) was used to compare attacker, midfielder, and defender position groups. Total distance, high-speed distance, and sprint distance were 9486 ± 300 m, 1014 ± 118 m, and 428 ± 70 m, respectively. Furthermore, attackers were observed to cover the greatest distance at all speeds compared to midfielders and defenders. Our findings suggest that the physical demands of Division I women’s soccer differ by position and appear lower compared to higher standards of play. Therefore, coaches and sports scientists responsible for the physical training of Division I collegiate players should consider the specific physical demands of the collegiate level and playing position when prescribing training, as well as in the development of their annual training programs.

## 1. Introduction

Investigations concerning the physical demands of women’s college soccer have spurred growing interest as the popularity of both soccer and women’s sports have increased. However, even with a growing interest in sport science and performance, a thorough understanding of the physical demands of women’s college soccer remains to be extensively explored. In general, research into women’s soccer has been scarce [1,2,3,4,5,6,7,8,9,10,11,12] leaving gaps in the literature regarding depth and breadth of information detailing the physical demands at different standards of play, particularly at the collegiate level [13]. In addition, many of the studies to date which have been published have been completed using small sample sizes with regard to number of players, number of matches, or both [4,6,11,14,15]. This being the case, further investigations into the physical demands of women’s soccer are warranted.

There is a disproportionate amount of literature relating to male players in comparison to female players. This is largely due to the cost associated with tracking and monitoring technology, which is often limited to professional men’s teams [12]. Though many aspects of men’s and women’s soccer are the same, such as pitch dimensions, match duration, number of players, goal size, and ball size, males and females can differ dramatically with regard to physical performance characteristics, with male players performing 30% more high-intensity activities during matches [14] and demonstrating superior performance across a range of fitness assessments [16,17,18]. As a result, research findings from observations of male players may not accurately reflect the physical demands of female players. There has been literature detailing physical demands of women’s soccer at the professional and international levels in which players have been observed to cover approximately 10,000 m per match, during which they perform between 70–190 high-intensity actions resulting in high-intensity running (HIR) distances ranging from 1530–1680 m and sprinting between 380–460 m [4,14,19]. Considering that HIR and sprinting efforts, as well as distance covered, have been shown to increase with the standard of play, it can be hypothesized that the activity profiles of women’s college soccer deviate from those found at the professional and international standard of play. Vescovi et al. [13] were the first and, to the authors knowledge, only published study to report on the locomotor characteristics of Division I women’s college soccer matches, finding players covered a mean total distance of 9930 m with 1080 m at high-speed (>15.5 km/h) and 267 m sprinting (>20 km/h). Considering much of the current literature objectifying activity profiles for women’s soccer have been investigations of professional and international level players [4,14,20,21,22] and with scant evidence existing to detail the physical demands of women’s soccer at the collegiate level, further investigation into this specific area of research is overdue. Therefore, the purpose of this study is to add to our understanding of the physical demands of women’s soccer matches at the Division I collegiate level, specifically the differences from higher standards of play and between playing positions, which is necessary for coaches and sports scientists to more appropriately prescribe training to maximize performance and minimize injury risks.

## 2. Materials and Methods

### 2.1. Experimental Approach to the Problem

For our study, we observed the match demands of women’s collegiate soccer players over four consecutive seasons to determine the physical demands associated with the Division I college level. Furthermore, playing positions were used to identify differing match requirements between defenders, midfielders, and attacking players. Global Positioning System (GPS) devices (Catapult Sports, Melbourne, Australia) sampling at 10 Hz were used to track player movements during competition. Units were secured to subjects using custom designed harnesses placing the GPS monitor on the upper-back between the shoulder blades. In accordance with previous investigations of female soccer players, velocity thresholds used to categorize player movements were: standing (0–0.1 km/h), walking (0.1–6.0 km/h), jogging (6.1–8.0 km/h), low-speed running (8.1–12.0 km/h), moderate-speed running (12.1–15.0 km/h), high-speed running (15.1–18.0 km/h), sprinting (18.1–25.0 km/h), and max sprinting (>25 km/h) [2,4,23]. Having been previously established as physical variables associated with match performance, total distance, high-speed running distance (>15 km/h), and sprint distances (>18 km/h) were chosen to be analyzed [2,4,23,24].

### 2.2. Subjects and Match Analysis

Unlike Vescovi et al. [13], who observed players from 9 NCAA institutions in single matches and did not require full-match participation, we observed twenty-three female college players with a mean age (years): 20.6 ± 1.0, body mass (kg): 62.1 ± 7.1, height (cm): 163.5 ± 13.3, and body fat (%): 22.4 ± 5.4 over four consecutive seasons providing 375 match observations. Our study was approved by an Institutional Review Board (IRB) on 20 February 2019 (c0219.11sw) and the subjects were informed of the benefits and risks of the investigation prior to signing an institutionally approved informed consent document to participate in the study. Matches were included only if players participated in the match in its entirety, without substitution. Additionally, all matches were contested in the U.S., on pitches meeting the established NCAA regulation with regard to field dimensions. In accordance with National Collegiate Athletic Association (NCAA) rules, matches consisted of two 45-min halves separated by a 15-min half time period. In the event of a tie, two 10-min, golden goal, extra time periods separated by a 2-min intermission were played. Players were categorized into one of three position groups for analysis: defenders, midfielders, or attackers. After collection, player data was downloaded and analyzed using the manufacturers proprietary software (Catapult OpenField & Catapult Sprint, Melbourne, Australia).

### 2.3. Match Activities

This investigation focused primarily on variables which objectify the physical demands of soccer match play, specifically total distance covered, high-speed distance, and sprint distance, narrowing our investigation to the most pertinent variables affecting match outcomes as well as those which are most easily understood by coaches and commonly explored by researchers. Within the current literature, there exists ample support for our focus on total distance, high-speed running distance, and sprint distance. Quantifying total distance provides an estimate of total running volume, additionally, it has historically been one of the most common variables studied [25]. According to Bangsbo [26], although the majority of distance is covered at low-speeds, periods of high intensity running are crucial to the outcomes of football matches by directly impacting goal-scoring opportunities [11]. Furthermore, the amount of high-speed running has been found to distinguish top-class players from those at lower levels [1,2,6], with top-class players covering 28% and 58% more high-speed running and sprint distances, respectively [24].

### 2.4. Statistical Analysis

The statistical software R (version 3.5.1) was used for all analyses. Linear mixed models fitted with maximum likelihood estimation were constructed for each dependent variable (total distance, high-speed running (HSR) distance, and sprint distance) using the lme4 package (version 1.1-20) [27,28]. Player position was treated as a fixed effect, while matches and athletes were included as crossed random intercept effects. The emmeans package (version 1.3.3) was used to compute both overall and position-specific means and to perform post hoc pairwise comparisons between positions; the resultant pairwise comparisons’ t-ratios were used to calculate Cohen’s d effect sizes via the psych package (version 1.8.12) [29]. Statistical significance was set at *p* ≤ 0.05. Effect size magnitudes are described according to Hopkins [30]: < 0.2 = trivial, 0.2–0.6 = small, 0.6–1.2 = moderate, and 1.2–2.0 = large. Data are presented as the estimate and associated 95% confidence interval.

## 3. Results

### 3.1. General Physical Demands

Total distance covered, high-speed running distance, and sprint distance covered by the sample at the Division I level, independent of playing position, are outlined in Table 1. Of the total distance covered, high-speed running distance (>15 km/h) per match accounted for approximately 10.7% of total match distance. Sprint distance values (>18 km/h), independent of playing position, were observed to be 428 ± 70 m with a mean of 31.1 ± 13 sprint efforts per match. Sprint distance represented 4.5% of total match distance and 42.2% of all high-speed running distance.

### 3.2. Position Specific Physical Demands

No statistically significant differences were observed for total distance covered between position groups for attackers and midfielders or between midfielders and defenders (*p* ≤ 0.05; Table 2). Values for total match distance covered by attackers were significantly greater than defenders (1333 m [*p* = 0.045], ES = 0.39 [-0.38–1.15], small). Similarly, attacker positions covered significantly more high-speed running distance compared to other position groups (attacker–midfielder: 493 m [*p* = 0.0035], ES = 1.38 [0.53–2.22], large; attacker–defender: 465 m [*p* = 0.0047], ES = 1.40 [0.51–2.27], large). No differences were observed for high-speed running distances between the midfield and defender position groups. Sprint distance values were observed to be statistically significant for attackers compared to midfielder positions (attacker–midfielder: 367 m [*p* = 0.003], ES = 1.73 [0.84–2.61], large) as well as for attackers compared to defender positions (attacker–defender: 249 m [*p* = 0.01], ES = 1.27 [0.40–2.12], large). No statistically significant difference was found between midfield and defender positions for sprint distance.

## 4. Discussion

The purpose of this study was to provide a thorough understanding of the activity profiles of women’s soccer at the Division I collegiate level and observe differences in positional demands and compared to higher standards of competition. Such information may provide evidence for coaches and sports scientists to more specifically target physical training to meet the demands of women’s collegiate soccer and the unique positional demands for attackers, midfielders, and defenders.

Our findings support the likelihood that the physical demands of women’s soccer at Division I collegiate level are lower in comparison to professional and international standards of play. These findings agree with those of Mohr [14] in that physical demands are specific to the standard of play, with higher standards of play typically requiring greater physical outputs. Total distance covered at the Division I collegiate level differs from the international level. Although some overlap is apparent, a total distance of 9486 m (9186–9786 m) covered by college females appears lower than international and professional players from Sweden [4,14,19], Denmark [2,4,14,19], Brazil [22], Australia [12,19], USA [14,19], and other European countries [11,19,31] which ranged from 9630–10,750 m. As a result, we can speculate that the lower physical demands of women’s soccer at the collegiate level are likely inadequate to prepare players for international competition.

Similarly, the match demands for high-speed running, 1014 m (895–1132 m), and sprinting, 428 m (359–498 m) were demonstrated to be less for college women at the Division I level compared to both professional and international level players. Although various measurement techniques have been used and velocity thresholds are highly inconsistent with regard to determining entry into high-speed running and sprinting zones, high-speed running and sprint volumes appear to be greater in professional and international level matches. Krustrup [2], Mohr [14], Andersson [4], and Bradley [7] found international and professional players to cover 1300–1680 m of high-intensity distances. Regarding sprint distance, it can be inferred that players at higher standards of play accumulate substantially higher sprint volumes. Given that our sprinting threshold has been set at >18 km/h, Mohr [14] observed professional and international players to sprint 380 m and 460 m respectively when sprint thresholds were set at >25 km/h. Such is also the case in a study by Andersson [4] which observed Scandinavian players to cover 221–256 m above the 25 km/h threshold. Given this evidence, it is reasonable to infer differences in sprint distances would have been much greater had velocity thresholds for sprinting been >18 km/h rather than >25 km/h.

High-intensity demands are particularly important for collegiate players also serving as members of their respective national teams. Unlike men’s soccer, it is common for international level players to also participate in Division I collegiate soccer. Players in such a role are often required to participate in national team camps and matches during the college season. These call-ups can expose players to acute spikes in high-intensity demands at the international level [13]. With this understanding, our data along with that of others [13,15] suggest that structuring training to develop greater capacities for high-intensity work through increased volumes of high-speed running and sprinting. As such, having an objective understanding of the differing physical demands between standards of play can provide useful information to ensure such players are supplemented with the appropriate volume and intensity of training required to bridge the performance gap between the collegiate and international level.

Positional differences have been found to exist at all standards of play between attackers, midfielders, and defenders. Traditionally, midfield players have been found to cover the greatest amount of total distance per match compared to attackers and defenders at the professional and international level [2,4,14]. In contrast, the findings of the current study observed attackers to cover the greatest volume of total distance, as well as high-intensity distance. Interestingly, Vescovi [13] reported similar findings for Division I college females with attackers covering the greatest total distance and high-intensity distance, although values for each variable were found to be higher for each position group compared to our results. This may be indicative of differing team tactical demands or collegiate soccer demands compared to professional and international standard of play. Nevertheless, although defenders, midfielders, and attackers exhibited similarities in position specific physical demands, distinct differences were observed between each of the three positional categories which can be used as a means for sport scientists and coaches to provide more specific training according to playing position.

It has been demonstrated that the physical demands of women’s soccer increase in a linear fashion as the standard of play advances from youth to college and beyond college to professional and international levels [2,4,13,14]. Due to these differences, coaches must consider the training history and physical capacity of their players. In general, players in the collegiate realm will not possess the same physical capacities as more elite players progressing to the professional and international level. Therefore, logic (and physiology) would suggest that it is inappropriate for collegiate players to be subjected to workloads suited for more elite players at higher levels. Such exposure to high workloads has the real possibility to result in non-functional overreaching or overtraining [31,32] and possibly increasing their risk of overuse injuries [33]. As a result, the addition of objective evidence to better understand the physical demands of women’s collegiate soccer at the Division I level to support the findings of Vescovi [13] and McCormack [15] are needed.

Some limitations to the present study include the use of a single Division I college women’s team. Although data was collected over the span of four seasons, it may not be fully representative of all Division I women’s soccer teams. Additionally, consideration must be given to the coach’s tactical finesse-oriented philosophy as well as player turnover and its effect on team composition across seasons. Furthermore, coaches’ philosophy and tactical strategy have been shown to influence match demands across various position groups [26]. This being the case, future studies would be well served to consider the inclusion of tactical formation to provide information regarding its effect on the physical demands. As a supplement to physical and tactical data, the addition of contextual match factors such as level of opponent, home vs. away, opponent’s tactical strategy, score, match result, and match significance would provide further useful information for coaches and practitioners. Lastly, further exploration of physical variables such as number, distance, and intensity of sprints, accelerations and decelerations, changes of direction, and jumps in addition to various post-match physiological and performance assessments to monitor post-match fatigue would provide a richer depiction of the activity profiles and resulting response of Division I collegiate players to competitive matches.

## 5. Conclusions

This study explored the physical demands of women’s soccer at the Division I level and the differences between position groups. Unlike Vescovi [13] who observed players from nine NCAA institutions in single matches and did not require full-match participation, we observed the physical demands for 23 athletes across four consecutive seasons, for which only full-match participation was required. Our findings provide objective data regarding the general and position-specific physical demands of women’s college soccer at the Division I level and can be useful for coaches and sports scientists to consider when constructing training and conditioning programs. Additionally, we have provided further evidence suggesting the physical demands of women’s soccer at the Division I level differ for total distance, high-speed distance, and sprint distance compared to higher standards of play and vary by position group between attackers, midfielders, and defenders. Therefore, coaches and sports scientists responsible for the physical training of Division I collegiate players must exercise a degree of caution when using published findings pertaining to the physical demands of professional and international level players as benchmarks for their sub-elite athletes.

## Figures and Tables

**Table 1 jfmk-04-00073-t001:** General physical demands of Division I women’s soccer.

Variables	Mean ± SD (m)	95% CI (m)	Velocity Threshold
TD	9486 ± 300	9186–9786	
HSRD	1014 ± 118	895–1132	>15 km/h
SPRTD	428 ± 70	359–498	>18 km/h

TD, Total distance; HSRD, High-speed running distance; SPRTD, Sprint distance; 95% CI, 95% Confidence Interval.

**Table 2 jfmk-04-00073-t002:** Position-dependent physical demands of Division I women’s soccer.

Variable	Attacker	Midfielder	Defender	Comparison *
TD	9882 (9414–10,349)	9536 (8998–10,034)	9039 (8527–9551)	A > D; A = M; M = D
HSRD	1333 (1147–1519)	840 (626–1054)	868 (665–1071)	A > D,M; M = D
SPRTD	633 (524–743)	267 (141–393)	385 (265–504)	A > D,M; M = D

Values presented as means (95% CI), m; TD, Total distance; HSRD, High-speed running distance; SPRTD, Sprint distance. * >, statistically significant difference; =, non-statistically significant difference.

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
