# Peer review of "The Physical Demands of NCAA Division I Women’s College Soccer"

_jfmk, 2019, doi:10.3390/jfmk4040073_

Round 1

Reviewer 1 Report

While novel in idea, I think it goes without saying that this is the type of information needed for the "GPS" Sport Science community as whole. 

While I am not familiar with the program at ETSU, I think it would be worth noting more the tactical side of the team used for this study (future study?). With the new hires with the USMNT, USWNT, and the USYNT the style of play or tactical set up is very much indicative of the physiological demands of matches and there has been a big shift the last three years in American style of play. Those in a soccer specific setting would find a little more detail relating the physiological demands to tactical set up extremely beneficial. 

All in all, well done. Good information that is useful. 

Author Response

Reviewer 1 feedback:

While I am not familiar with the program at ETSU, I think it would be worth noting more the tactical side of the team used for this study (future study?). With the new hires with the USMNT, USWNT, and the USYNT the style of play or tactical set up is very much indicative of the physiological demands of matches and there has been a big shift the last three years in American style of play. Those in a soccer specific setting would find a little more detail relating the physiological demands to tactical set up extremely beneficial. 

In response to the feedback from reviewer 1, we added a sentence for future direction to state the more tactical finesse-oriented philosophy of the soccer coach. Additionally, suggestions for the inclusion of most specific tactical information, as well as contextual match factors were recommended to better understand how philosophy and formation may affect physical demands.

Reviewer 2 Report

General: My congratulations to the authors, I found the article interesting and insightful, and particularly well written. I think the authors have tried to put their findings into context and my comments are for clarifications and suggestions.

Title: The title is specific and concise about what the paper is about.

Abstract: The authors must rewrite the abstract. 1) Line 11-25:  Authors should consider changes in the abstract of the paper. The abstract should be a total of about 200 words maximum. The abstract should be a single paragraph and should follow the style of structured abstracts, but without headings: 1) Background: Place the question addressed in a broad context and highlight the purpose of the study; 2) Methods: Describe briefly the main methods or treatments applied. Include any relevant preregistration numbers, and species and strains of any animals used. 3) Results: Summarize the article's main findings; and 4) Conclusion: Indicate the main conclusions or interpretations.

Keywords: Authors should consider changes in the keywords of the paper. For example: College Soccer; GPS; Physical Demands; Women’s soccer; Physical match performance

Introduction:  No comments, well written.

Materials and Methods: 1) Line 78-80: In relation to the purpose of the study (understanding the physical demands of womens’ soccer at the collegiate level), the inclusion of the variables number of sprints and accelerations and decelerations would make the work more complete and richer. Is not possible to include this variables?

2) Line 88: NCAA (National Collegiate Athletic Association)

Results: No comments, well presented.

Discussion: Generally well written. The authors should refer future research directions, for example: post-match assessments of physical aspects of woman’s soccer performance can be made more objective by factoring in the effects of situational variables.

Conclusions: The authors must rewrite this section. 1) The limitations described (Line 224-227) should  be included in Discussion section

2) Line 211-214: “Unlike Vescovi…..was required.”…should  be included in Subjects and Match Analysis section

References: Authors should correct this section.

References must be numbered in order of appearance in the text (including table captions and figure legends) and listed individually at the end of the manuscript.

In the text, reference numbers should be placed in square brackets [ ], and placed before the punctuation; for example [1], [1–3] or [1,3]. For embedded citations in the text with pagination, use both parentheses and brackets to indicate the reference number and page numbers; for example [5] (p. 10). or [6] (pp. 101–105).

References should be described as follows, depending on the type of work:

  Journal Articles:
1. Author 1, A.B.; Author 2, C.D. Title of the article. Abbreviated Journal Name Year, Volume, page range.

Author Response

Reviewer 2 feedback:

General: My congratulations to the authors, I found the article interesting and insightful, and particularly well written. I think the authors have tried to put their findings into context and my comments are for clarifications and suggestions.

Title: The title is specific and concise about what the paper is about.

Abstract: The authors must rewrite the abstract. 1) Line 11-25: Authors should consider changes in the abstract of the paper. The abstract should be a total of about 200 words maximum. The abstract should be a single paragraph and should follow the style of structured abstracts, but without headings: 1) Background: Place the question addressed in a broad context and highlight the purpose of the study; 2) Methods: Describe briefly the main methods or treatments applied. Include any relevant preregistration numbers, and species and strains of any animals used. 3) Results: Summarize the article's main findings; and 4) Conclusion: Indicate the main conclusions or interpretations.

Abstract was revised to better align with the journal format based on suggestions provided.

Keywords: Authors should consider changes in the keywords of the paper. For example: College Soccer; GPS; Physical Demands; Women’s soccer; Physical match performance

Keywords were revised according to the recommendations of reviewer 2.

Introduction: No comments, well written.

Materials and Methods: 1) Line 78-80: In relation to the purpose of the study (understanding the physical demands of womens’ soccer at the collegiate level), the inclusion of the variables number of sprints and accelerations and decelerations would make the work more complete and richer. Is not possible to include this variables?

The purpose statement was revised to de-emphasize the difference between competitive level and speak more to elucidating the physical demands of the Division I collegiate level. Additionally, an estimation of the number of sprint efforts was included, however, a more detailed analysis of accelerometer data and sprint efforts were not included due to being prepared in a second manuscript.

2) Line 88: NCAA (National Collegiate Athletic Association)

Results: No comments, well presented.

Discussion: Generally well written. The authors should refer future research directions, for example: post-match assessments of physical aspects of woman’s soccer performance can be made more objective by factoring in the effects of situational variables.

Suggestions to recommend the inclusion of post-match assessments of physical aspects to objectify match effect in future research has been added.

Conclusions: The authors must rewrite this section. 1) The limitations described (Line 224-227) should be included in Discussion section

2) Line 211-214: “Unlike Vescovi…..was required.”…should be included in Subjects and Match Analysis section

Limitations and future direction has been removed from the conclusion and added to the discussion section per recommendation.

The line references above beginning with “Unlike Vescovi…..was required” was added to the subjects and match analysis section. However, the authors feel it adds to the conclusion, therefore, we did not remove it from this section.

References: Authors should correct this section.

References must be numbered in order of appearance in the text (including table captions and figure legends) and listed individually at the end of the manuscript.

In the text, reference numbers should be placed in square brackets [ ], and placed before the punctuation; for example [1], [1–3] or [1,3]. For embedded citations in the text with pagination, use both parentheses and brackets to indicate the reference number and page numbers; for example [5] (p. 10). or [6] (pp. 101–105).

References should be described as follows, depending on the type of work:

 Journal Articles:

Author 1, A.B.; Author 2, C.D. Title of the article. Abbreviated Journal Name Year, Volume, page range.

Our references have been formatted according to Microsoft EndNote, MDPI format. Upon review, reference are listed in order of appearance.

Reviewer 3 Report

I have read the paper and I have found it as interesting and well written one. The authors describe match activity profiles of female soccer players representing Division I collegiate level. Based on linear mixed modelling (well used for this purpose) the authors compared activity structure of players at different positions.
The aim of the study: I would rather suggest omitting the differnces from higher standards of play (as a purpose). This should be of course widely discussed (as it has been done), but this comparision is based on other studies.

Minor remarks:
Units should be written after space: '230 m' rather than '230m'.
In Results part there are some numbers which have been presented twice: In the text and in tables 1 and 2. If you decided to use tables, do not put numbers in the text. You can just comment that some mean values were greater or less and variables had different structure (profile), etc.
I do not like the way of citing like in line #157: 8 positions at once.

Author Response

Reviewer 3 feedback:

I have read the paper and I have found it as interesting and well written one. The authors describe match activity profiles of female soccer players representing Division I collegiate level. Based on linear mixed modelling (well used for this purpose) the authors compared activity structure of players at different positions.

The aim of the study: I would rather suggest omitting the differnces from higher standards of play (as a purpose). This should be of course widely discussed (as it has been done), but this comparision is based on other studies.

The authors have revised the purpose to focus on elucidating the physical demands of Division I collegiate women’s soccer and the differences between positions.

Minor remarks:

Units should be written after space: '230 m' rather than '230m'.

Corrections to units have been made accordingly

In Results part there are some numbers which have been presented twice: In the text and in tables 1 and 2. If you decided to use tables, do not put numbers in the text. You can just comment that some mean values were greater or less and variables had different structure (profile), etc.

The results section has been revised as suggested.

I do not like the way of citing like in line #157: 8 positions at once.

This has been revised to remove the string of 8 references, and cite reference corresponding to specific countries.